# Evaluation of Label-Free Confocal Raman Microspectroscopy for Monitoring Oxidative Stress In Vitro in Live Human Cancer Cells

**DOI:** 10.3390/antiox11030573

**Published:** 2022-03-17

**Authors:** Jakub Maciej Surmacki, Isabel Quiros-Gonzalez, Sarah Elizabeth Bohndiek

**Affiliations:** 1Department of Physics, University of Cambridge, JJ Thomson Avenue, Cambridge CB3 0HE, UK; quirosisabel@uniovi.es; 2Cancer Research UK Cambridge Institute, University of Cambridge, Robinson Way, Cambridge CB2 0RE, UK; 3Institute of Applied Radiation Chemistry, Lodz University of Technology, Wroblewskiego 15, 93-590 Lodz, Poland; 4Animal Histopathology Core at IUOPA, University of Oviedo, 33006 Oviedo, Spain; 5Redox Biology and Metabolism in Cancer, Instituto de Investigación Biosanitaria ISPA, 33006 Oviedo, Spain

**Keywords:** oxidative stress, Raman microspectroscopy, fluorescence microscopy, tert-butyl hydroperoxide, *N*-acetyl-l-cysteine

## Abstract

Understanding the impact of free radicals and antioxidants in cell biology is vital; however, noninvasive nonperturbative imaging of oxidative stress remains a challenge. Here, we evaluated the ability of label-free Raman spectroscopy to monitor redox biochemical changes in antioxidant (*N*-acetyl-l-cysteine, NAC) and pro-oxidant (tert-butyl hydroperoxide, TBHP) environments. Cellular changes were compared to fluorescence microscopy using CellROX Orange as a marker of oxidative stress. We also investigated the influence of cell media with and without serum. Incubation of cells with NAC increased the Raman signal at 498 cm^−1^ from S-S disulphide stretching mode, one of the most important redox-related sensors. Exposure of cells to TBHP resulted in decreased Raman spectral signals from DNA/proteins and lipids (at 784, 1094, 1003, 1606, 1658 and 718, 1264, 1301, 1440, 1746 cm^−1^). Using partial least squares–discriminant analysis, we showed that Raman spectroscopy can achieve sensitivity up to 96.7%, 94.8% and 91.6% for control, NAC and TBHP conditions, respectively, with specificity of up to 93.5, 90.1% and 87.9%. Our results indicate that Raman spectroscopy can directly measure the effect of NAC antioxidants and accurately characterize the intracellular conditions associated with TBHP-induced oxidative stress, including lipid peroxidation and DNA damage.

## 1. Introduction

Free radicals, including reactive oxygen species (ROS), are generated as a normal by-product of respiration, and at low concentrations, they act as signalling molecules and are essential for normal cell and tissue homeostasis [1,2]. Under normal physiological conditions, a delicate balance between the rate of ROS generation and the activity of antioxidant systems is maintained [3,4]. Under pathological conditions, this balance is often disrupted, which can lead to changes in biological function [3,5]. Oxidative stress, and eventually cell death, occurs when the concentration of free radicals exceeds the capacity of the intracellular antioxidant systems [6]; this plays a key role in the progression of a range of pathologies [4,7,8], including many aspects of cancer initiation and development [9].

For example, cancers of the gastrointestinal tract and lung frequently arise from sites of chronic inflammation [10,11], where the sustained oxidative environment can damage healthy epithelial and stromal cells and lead to malignant transformation [11]. Once transformed, the aberrant metabolism and proliferation of cancer cells, combined with an inadequate neovasculature and inflammatory cell infiltration, leads to even higher levels of ROS [12]. Cancer cells must therefore tightly regulate their antioxidant capacity to ensure that they can survive this continual ROS exposure. The ability to endure both prolonged and severe oxidative stress has been strongly associated with more aggressive disease and the emergence of drug resistance [13].

Our ability to study the role of free radicals and antioxidants in biology is limited by the scarcity of methods to interrogate these dynamic systems in living systems, from cell cultures to intact organisms. It is a major deficiency in the field of molecular imaging and has profoundly limited research in redox biology. Standard approaches to measure redox state, such as high-performance liquid chromatography (HPLC), immunohistochemistry [4] or electrochemical methods [14], require cells or tissues to be excised and disrupted. Current methods for the study of live cells typically involves optical measurement, using either fluorescence microscopy or flow cytometry together with an activatable fluorescent dye [15,16,17] or genetically encoded redox-sensitive fluorescent protein [10,18]. These methods can also be extended for application in living organisms such as mice using bioluminescence [19] and photoacoustic [20] imaging; however, all these cases require the introduction of an exogenous agent to provide the measured contrast, which has the potential to perturb the system of interest. 

Raman spectroscopy is a powerful bioanalytical technique that directly reveals the chemical constituents of a given sample based on the inelastic scattering properties of molecular bonds. Despite the relatively weak nature of the Raman effect, the advent of confocal Raman microspectroscopy methods that allow 3D localization of signals together with highly sensitive detectors has enabled this label-free technique to be applied in living cells [21,22,23]. Based on our prior preliminary study [24], we hypothesized that chemical modifications produced by oxidative stress in cancer would alter the Raman spectral signature, providing insight into redox state variation. To test this hypothesis, we applied confocal Raman microspectroscopy to human lung carcinoma cells exposed to *N*-acetyl-l-cysteine (NAC, antioxidant) and tert-butyl hydroperoxide (TBHP, pro-oxidant). We demonstrate here that Raman spectroscopy is able to detect the effects of both the antioxidant and pro-oxidant on live cells in culture.

## 2. Materials and Methods

### 2.1. Cell Culture and Preparation for Microspectroscopy

A human Caucasian lung carcinoma A549 (from ATCC) cell line was used for all studies. The A549 cell line was grown in DMEM/F-12 (Gibco, Life Technologies Billings, MT, USA, cat. no. 11039-021) supplemented with 10% foetal bovine serum (Gibco, Life Technologies, Billings, MT, USA, cat. no. 16000-044). Cells were maintained at 37 °C in humidified atmosphere containing 5% CO_2_. Cells for microspectroscopy analysis were seeded in a 6-well plate with a 25 mm round quartz coverslip (UQG Optics, Cambridge, UK, cat. no. CFQ-2520) in phenol-red-free medium at a density of 2 × 10^5^ cells per 35 mm well and left overnight to adhere. Before Raman measurements, quartz coverslips were mounted into an Attofluor cell chamber (Invitrogen, Waltham, MD, USA, cat. no. A-7816). Cells were washed with phosphate-buffered saline to remove any unattached cells, and fresh medium was added. Experiments were thus performed in fresh media, either with or without serum. No antibiotics were used in cell culture. The cell line was confirmed to be free of mycoplasma contamination using MycoProbe^®^ Mycoplasma Detection Kit (R&D Systems, Minneapolis, MN, USA).

Raman measurements were performed after 1 h of treatment with either 1 mM *N*-acetyl-l-cysteine (NAC, Sigma-Aldrich, Merck Life Science UK Limited, Gillingham, UK, cat. no. A7250) or 200 μM tert-butyl hydroperoxide (TBHP, cat. no. 488139 Aldrich, Merck Life Science UK Limited, Gillingham, UK). NAC can have direct antioxidant activity but is also a precursor of reduced glutathione, a power intracellular antioxidant and substrate of several antioxidants [25]. TBHP is an organic hydroperoxide that is commonly used as a model ROS inducer for evaluation of mechanisms of cellular alterations resulting from oxidative stress in cells and tissues. The cytotoxic effects exerted by TBHP include: GSH depletion [26]; lipid peroxidation [26,27,28]; permeabilization of cell [29] and mitochondrial [30] membranes; impaired ATP synthesis [26]; and DNA damage [31]. Furthermore, as mentioned above, the experiments were performed both with and without serum, as cell culture serum is known to provide antioxidant capacity and be an additional stress condition for testing [32]. Solutions of NAC and TBHP were prepared in the culture media of the cells, DMEM/F-12, which contained 15 mM HEPES buffer to maintain pH and has been shown to be optimal for studies of serum-free media as it compensates for the loss of buffering capacity under serum starvation.

### 2.2. Fluorescence Data Acquisition and Analysis

The experimental conditions used to generate reduction/oxidative activity were established using live fluorescence imaging analysis with CellROX Orange Reagent for oxidative stress detection (cat. no. C10443, ThemoFisher Scientific, Waltham, MD, USA). The nonfluorescent CellROX Orange becomes fluorescent in the presence of a wide variety of reactive oxygen species, including peroxyl (ROO^●^) and hydroxyl (HO^●^) radicals [33].

Live fluorescence imaging was performed with an Olympus FV1200 laser scanning confocal microscope containing a PMT detector using 60× oil immersion objective equipped with 405, 440, 473, 488, 514, 559 and 635 nm diode lasers and CellVivo controllers (temperature: 37 °C, humidity: 100%, CO_2_: 5%). A549 cells were seeded (5 × 10^5^ cells/3 mL) on µ-slides (8 well, ibiTreat, 80826) and sustained in DMEM/F-12 with 10% FBS at 37 °C and 5% CO_2_. Following treatment with the experimental conditions for Raman spectroscopy (Control, NAC or TBHP; with and without serum), each well was incubated for 30 min at 37 °C in the optimal concentration of the probe for staining (6 μL of 250 μM CellROX Orange dye in dimethyl sulfoxide dissolved in 3 mL of warm medium). The staining solution was removed, and cells were washed 3 times with warmed medium before imaging. Images (512 × 512 pixels) were recorded with: scan speed of 10μs/pixel; confocal aperture at 105 μm; and excitation: 559 nm laser. Fluorescence emissions of CellROX Orange Reagent for oxidative stress detection were measured at 570–670 nm. Fluorescence images were analyzed using ImageJ.

### 2.3. Raman Data Acquisition and Analysis

Raman spectroscopy was performed with a confocal inverted Raman microscope (Alpha 300M+, WITec) equipped with a 785 nm single-mode diode laser (XTRA II; Toptica Photonics Inc., Farmington, NY, USA), a 300 mm triple-grating imaging spectrometer (Acton SpectraPro SP-2300; Princeton Instruments Inc., Trenton, NJ, USA) with 600 g/mm grating (BLZ = 750 nm), a thermoelectrically cooled CCD camera (DU401A-BV; Andor, Ireland), a 60× water immersion objective (Nikon CFI Achromat Flat Field 60×, NA 0.80 WD 0.3, MRP00602) and Digital Pixel Imaging System for temperature, CO_2_ level and humidity control. The spectral resolution of the Raman data, dispersed by a 300 mm focal length monochromator incorporating a 600 g/mm grating, varied between 3 and 5 cm^−1^. The Raman system was calibrated by HG-1 Mercury Argon Calibration Source from Ocean Optics (Hg and Ar Lines from 253–922 nm). Raman peak positions were confirmed using a silicon wafer and polystyrene as references samples.

Single Raman spectra were collected with 1 s integration time and 30 accumulations. Total numbers of recorded Raman spectra used in the analysis were: control (with serum, n = 610 from 61 live cells; without serum, n = 400 from 40 cells); NAC (with serum, n = 610 from 61 live cells; without serum: n = 400 from 40 cells); and TBHP (with serum, n = 810 from 81 live cells; without serum: n = 610 from 61 cells). Data were acquired from multiple independent biological experiments (control with serum, n = 3; control without serum, n = 2; NAC with serum, n = 3; NAC without serum, n = 2; TBHP with serum, n = 4; TBHP without serum, n = 3). Raman images with size area of 100 × 100 μm^2^ (160 × 160 points) were recorded with 0.3 s integration time at 785 nm excitation (120 mW power).

Data processing was performed using Project Plus Four (WITec GmbH, Ulm, Germany), Origin 2016 (OriginLab, Northampton, MD, USA) and MATLAB (Mathworks, Portola Valley, CA, USA) with PLS-Toolbox (Eigenvector Research Inc., Manson, WA, USA). All Raman spectra were cosmic ray and baseline corrected (polynomial order: 5) then smoothed with a Savitzky–Golay filter (order 3, 4pt). PCA and PLSDA were performed in MATLAB using the PLS-Toolbox. For PLSDA single-point-based data, the spectra were split into sets for calibration [n(control with serum) = 458, n(control without serum) = 300, n(NAC with serum) = 457, n(NAC without serum) = 300, n(TBHP with serum) = 608, n(TBHP without serum) = 458] and validation [n(control with serum) = 152, n(control without serum) = 100, n(NAC with serum) = 153, n(NAC w/o serum) = 100, n(TBHP with serum) = 202, n(TBHP without serum) = 152] by removing every fourth spectrum to form the validation set. Spectra were normalized to the area under curve. Cross validation was performed using venetian blinds and 10 data splits, and the model was built using 7 latent variables. Statistical significance was analyzed using Kruskal–Wallis ANOVA. *p* < 0.05 was considered statistically significant.

## 3. Results

### 3.1. Fluorescence Imaging Reveals Changes in the Levels of Oxidative Stress in A549 Cells with Antioxidant or Pro-Oxidant Treatment

We first investigated the effect of NAC and TBHP treatment in the A549 cells using the fluorescence probe CellROX Orange (Figure 1). The low level of fluorescence emission observed in control cells reduced slightly after treatment with NAC within 1 h (Figure 1A,B; control^1H^ 1.00 ± 0.10 a.u. vs. NAC^1H^ 0.71 ± 0.08 a.u.) and returned to baseline at the 2 h timepoint (Figure 1A,B; control^2H^ 0.98 ± 0.17 a.u. vs. NAC^2H^ 0.87 ± 0.12 a.u.). Conversely, the fluorescence emission was dramatically increased at 1 h after induction of oxidative stress upon treatment with TBHP (Figure 1C; control^1H^ 1.00 ± 0.10 a.u. vs. TBHP^1H^ 2.62 ± 0.42 a.u.) and sustained for 2 h (Figure 1C; control^2H^ 0.98 ± 0.17 a.u. vs. TBHP^2H^ 1.61 ± 0.34 a.u.). Cells deprived of the antioxidant capacity afforded by serum in their media showed almost two-fold higher fluorescence emissions compared to cells incubated in complete media (control without serum 9.04 ± 1.42 vs. control with serum 4.57 ± 0.46 a.u., *p* = 0.0150), which was further enhanced by the addition of TBHP (Figure 1D,E; 2.35 ± 0.61 a.u.). In both serum conditions, the anti- and pro-oxidant effects of NAC and TBHP became less clear by the 2 h timepoint; hence the 1 h timepoint was selected for future studies. These findings served to verify that the experimental conditions of 1mM NAC treatment and 200 μM TBHP generated the requisite changes in cellular oxidative stress after 1 h of incubation; hence we chose these conditions to interrogate using Raman spectroscopy.

### 3.2. Raman Microspectroscopy and Imaging Indicate the Major Vibrational Modes That Are Altered under Different Levels of Oxidative Stress

Raman spectra and imaging data were acquired in A549 cells in the control and treated conditions tested using fluorescence microscopy. We first looked at the median spectra obtained (Figure 2A), which allowed us to identify the major Raman vibrational modes (referred to as ‘bands’ for the remainder of the paper) present across all conditions at 498, 718, 784, 850, 1003, 1046, 1094, 1264, 1301, 1340, 1440, 1578, 1606, 1658 and 1746 cm^−1^. These bands correspond to major classes of essential macromolecules present in living organisms, such as nucleic acids, proteins, carbohydrates and lipids (Table 1). While several vibrational modes can often contribute to a single band (as noted in Table 1), the bands can be grouped according to their predominant origin: from disulphide stretch (498cm^−1^), nucleic acids (784 and 1094 cm^−1^), lipids/phospholipids (718, 1264, 1340, 1440, 1658 and 1746 cm^−1^) and proteins (1003, 1440, 1606 and 1658 cm^−1^).

The only difference in the identified Raman bands in the cells without serum was an additional peak at 880 cm^−1^, which was elevated in the TBHP condition without serum (Figure 2B). The Raman band at 880 cm^−1^ can be attributed to the indole ring mode of tryptophan, which involves both ring stretching and displacement of the imino group. The indole ring was sensitive to hydrogen bonding, with the mode shifting from 883 cm^−1^ (no hydrogen bonding) to 871 cm^−1^ (strong hydrogen bonding) [34]. This result points to an increase in the reduction by H^+^ of the indole ring. Given that the identified Raman bands are largely consistent between the media conditions with and without serum, for clarity, the data without serum will mostly be presented as supplementary data for the subsequent analyses unless any notable differences were observed between the conditions.

We next composed images from the Raman spectral data to reflect the major components of nucleic acids, lipids and proteins (Figure 3; without serum condition shown in Appendix A). Detailed inspection of presented Raman images indicates that membranous cell organelles such as the endoplasmic reticulum/the Golgi apparatus/mitochondria (observed in the perinuclear regions) and lipid droplets showed relatively high concentrations of lipids (CH_2_ and CH_3_ deformations, band at 1440 cm^−1^) as opposed to the nucleus region, which showed higher concentrations of DNA (ring breathing mode of pyrimidine bases, 784 cm^−1^), as would be expected. Regions with protein composition were indicated by Amide I (1658 cm^−1^) and the symmetrical stretching mode of the phenyl group from phenylalanine (1003 cm^−1^); these regions strongly overlap with the lipid and DNA regions. Cells deprived of serum and those treated with TBHP showed more focal lipid regions at the periphery, which was likely due to an accumulation of lipid droplets in response to oxidative stress and may be associated with the early events in the apoptotic cascade [40].

### 3.3. Principal Components Analysis Confirms the Discrimination Power of Raman Spectroscopy

For classification and identification purposes, we first employed a principal components analysis (PCA). PCA reduces the dimensionality of complex datasets while minimizing information loss [41], with the resulting scores showing the position of each observation in the new coordinate system of principal components and loadings indicating how much each variable contributes to a particular principal component. Applying PCA to our data enabled a detailed evaluation of the resolving power of Raman spectroscopy to discriminate between antioxidant and pro-oxidant conditions.

In the presence of serum, PCA was clearly able to distinguish between the control and NAC-treated conditions based on the derived principal components (Figure 4A; PC2−5), though the distinction of TBHP treatment was more subtle and best captured by a subset of the principal components (PC4 vs. PC2). These data illustrate the capability of Raman spectroscopy to detect differences in oxidative stress, akin to our findings with the CellROX Orange fluorescent probe (Figure 1), but with the benefit of being derived in a label-free manner and in addition providing detailed insight into the molecular changes in the cell induced by the antioxidant and pro-oxidant conditions. In cells without serum, the three conditions were more obviously distinguished in all score plots (Appendix A), most likely because the added stress induced by serum deprivation led to more substantial changes in their spectra.

Detailed inspection of the PCA loadings (Figure 4B), indicates the conditions are delineated based on changes in the Raman spectra in a subset of peaks, which suggests not only the propagative peroxidation of lipids under oxidative stress conditions (bands at 718, 1264, 1301, 1440 and 1746 cm^−1^) but also changes in proteins (1003, 1606 and 1658 cm^−1^) and DNA (784 and 1094 cm^−1^). Moreover, the Raman band at 498 cm^−1^ associated with S-S disulphide stretching is implicated in PC3-PC5, which would be expected given the important role of thiol groups in intracellular antioxidant balance, both in glutathione, a major intracellular antioxidant, and redox-sensitive proteins [42].

Based on these findings, we then applied a Kruskal–Wallis ANOVA test to further investigate the changes in the discriminating Raman bands of interest from the PCA loadings plots (Figure 4B). We used the Kruskal–Wallis test as our integral intensity signals did not meet the normality assumption of a one-way ANOVA. All bands analyzed were significantly different for all three classes—control, NAC and TBHP—at the 0.05 level (Appendix A).

As suggested from the PCA results, NAC was found to increase the intensity of the S-S disulphide stretch at 498 cm^−1^, indicating that excess electrons pass to thiols and disulphide bonds are formed, while THBP incubation or serum starvation significantly decrease the intensity, indicating a decrease in the S-S bond (Figure 5A). The cytosine, uracil, thymine and pyrimidine ring breathing modes in RNA and DNA (Figure 5B) showed a similar pattern, with an increase in intensity with NAC and a commensurate decrease in response to TBHP treatment in media with serum. The Raman bands associated with lipids/phospholipids showed a decreasing trend for both treatments (Figure 5C,D); however, in serum-free media, the decrease was more dramatic in NAC-treated compared to TBHP-treated cells. The Raman band associated with proteins through vibrational modes from phenylalanine, proline and the symmetric stretching mode of the phenyl group (Figure 5E) showed a similar behaviour to that in the lipid/phospholipid cases. In the remaining bands investigated, application of either treatment typically reduced the intensity of the Raman bands, with NAC treatment having the greater effect in media without serum and TBHP having the greater effect in media containing serum.

### 3.4. Partial Least Squares–Discriminant Analysis Shows Good Classification Performance for Antioxidant and Pro-Oxidant Conditions

Finally, we undertook a partial least squares–discriminant analysis (PLS-DA) to test whether the Raman spectroscopy data could be used to accurately discriminate between our antioxidant and pro-oxidant conditions. Cross-validation was performed to avoid overfitting. The calibration and validation data sets defined in Materials and Methods (Section 2.3) are indicated in Figure 6 (Appendix A without serum). Our classification results (Table 2) and associated confusion matrices (Appendix A) show that Raman spectroscopy can indeed discriminate between oxidative stress conditions in the A549 cell line within root mean square errors that are comparable with previous values discriminating between cell lines [23]. It is also interesting to note that the model is better able to determine NAC treatment than TBHP.

The resulting receiver operating characteristic curves of all Raman data (Figure 6D–F and Appendix A yielded areas under curves (AUC) of: 0.9924 (w/o serum = 0.9787), 0.9761 (w/o serum = 0.9993) and 0.9610 (w/o serum 0.9871) for the control, NAC and TBHP, respectively. The high values of ROC’s AUC confirm the ability of the test to correctly classify Raman spectra into three groups.

## 4. Discussion

Oxidative stress plays a key role in the progression of many pathological conditions, yet our ability to study the process in living systems currently requires the use of labelling, either using dyes or genetically encoded reporters, which may perturb the system of interest themselves. We investigated the potential of Raman microspectroscopy as a label-free tool to distinguish between cells under antioxidant and pro-oxidant conditions.

Taken together, our analyses reveal that Raman spectroscopy is sensitive to redox changes in different macromolecules in live cells. The Raman spectral data were also sufficient to enable classification of the different conditions with high sensitivity and specificity. More importantly, several Raman vibrational modes gave insight into the underlying biological processes that resulted from NAC or TBHP treatment.

Firstly, the Raman band at 498 cm^−1^ is associated with S-S disulphide stretching and was identified as a discriminating peak in our PCA analysis, showing a clear increase under NAC treatment and conversely a decrease with TBHP treatment. S-S disulphide stretching would be expected to be important for discriminating between these conditions, given the important role of thiol groups in intracellular antioxidant balance, both in the glutathione redox couple (GSSG-GSH), a major intracellular antioxidant system, and also in redox-sensitive proteins [42,43]. NAC may increase intracellular glutathione levels and also mediate thiol–disulfide exchange reactions, explaining the observed increase [44,45], while TBHP-treated cells may use the reducing capacity of GSSG and other thiol groups to combat the pro-oxidant activity of TBHP. Noninvasive assessment of thiol status in this manner could be of high value to the antioxidant research community. In fact, our results indicate that Raman has better performance in detecting antioxidant changes induced by NAC than the widely used CellROX Orange probe, which showed only subtle differences between the control and NAC groups.

Secondly, the Raman band at 784 cm^−1^ associated with nucleic acid ring breathing modes showed a similar pattern to the S-S disulphide stretching, suggesting a similar oxidative-stress-specific readout. It is well-known that free radicals can cause oxidative damage to RNA and DNA bases, which can be used as a biomarker of oxidative stress using ex vivo methods [46]. The decline in peak intensity with TBHP is consistent with prior work where Raman spectroscopy detected a similar reduction in peak intensity after induction of oxidative stress with hydroxyl radical treatment in neuronal cells [47].

We also noted the appearance of an additional Raman band at 880 cm^−1^ associated with tryptophan in the TBHP-treated cells without serum. The tryptophan degradation product quinolinic acid is an NAD^+^ precursor and has been implicated in the oxidative stress response in glioma [48], which may explain its presence in the extreme stress condition of TBHP without serum. Further work would be needed to examine this finding and determine whether it is indeed linked directly to redox changes or is a result of the broader cellular changes that occur in response to serum deprivation.

Nonetheless, despite promising findings, there are several limitations of our study and avenues of further work that should be explored to establish Raman spectroscopy as a tool in antioxidant biology. Firstly, the experimental conditions for our study were established in independent experiments using detection of ROS by confocal fluorescence microscopy rather than directly in the cell cultures where Raman spectroscopy was performed. Secondly, several of the discriminating Raman bands identified with PCA showed a general decrease with both treatments, suggesting that there are other processes beyond direct redox modifications that are occurring, though these may be inducible by redox-sensitive pathways. Thirdly, our imaging data suggest that some of our experimental conditions, such as those without serum, may have stimulated the first steps of apoptosis, according to the concentration of lipid droplets in the cells. While this would not be problematic in standard cell culture experiments complete with serum, it should be controlled for in future studies on serum deprivation with further protocol optimization.

Finally, the relatively weak nature of Raman scattering requires a trade-off between the signal-to-noise ratio and acquisition time, which limited the exploration of antioxidant dynamics in this study. Additionally, the Raman cross-section of many relevant biomolecules is too low to be confidently examined using spontaneous Raman spectroscopy. To overcome these limitations, it would be prudent to evaluate the use of coherent Raman spectroscopy methods to enhance the signal in future studies. It would also be interesting to use other laser excitation wavelengths to broaden the range of molecules that could be examined. For example, using 488 nm excitation enables resonant Raman excitation of carotenoids, an important antioxidant system, particularly in the eye and skin [49]. 532 nm excitation enables resonance Raman excitation of cytochrome c, a regulator of oxidative stress [50], which generates distinct Raman bands associated with its oxidized and reduced forms [51]. Together with our findings, it is possible that multiwavelength Raman excitation could therefore be used to inform on the intricate balance of a range of intracellular redox systems. Nonetheless, with the experiments presented here, we were able to visualize the dynamic response of cells to both antioxidant and pro-oxidant conditions using Raman spectroscopy.

## 5. Conclusions

The results obtained in this study confirm that Raman spectroscopy is a unique label-free technique that can classify the oxidative stress condition of live cells with high sensitivity and specificity. Moreover, Raman spectroscopy can report specifically on: disulphide stretching associated with thiol antioxidant systems; oxidative damage to RNA and DNA bases; and an elevation of tryptophan concentration in highly oxidizing environments. Raman spectroscopy could therefore be applied for in vitro redox biology research.

## Figures and Tables

**Figure 1 antioxidants-11-00573-f001:**
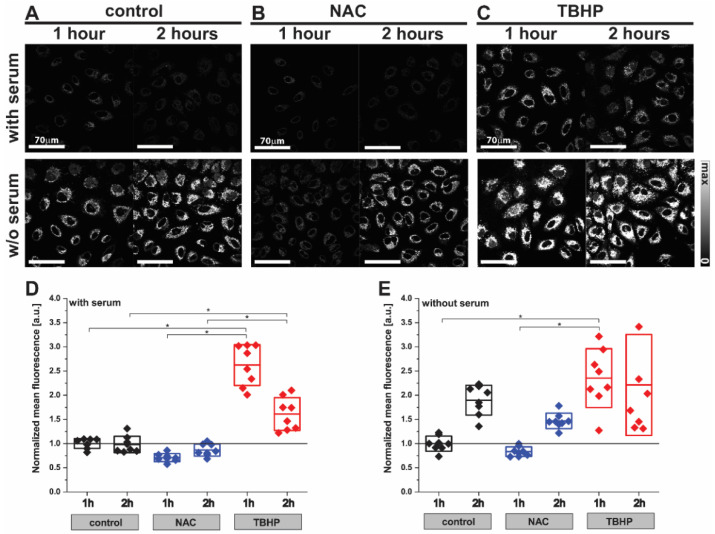
Detection of oxidative stress in live A549 cells using the CellROX Orange probe by fluorescence imaging. Images of A549 cells ((**A**) control) after treatment with (**B**) NAC (1 mM) or (**C**) TBHP (200 μM) for 1 or 2 h. Cells were labelled with CellROX Orange, which fluoresces when oxidized by ROS. Two-way ANOVA analysis was performed using the mean fluorescence intensity of TBHP- or NAC-treated A549 cells normalized to the control sample (**D**) with or (**E**) without serum. Box plots indicate means ± standard deviation; * *p* < 0.05 was considered as statistically significant.

**Figure 2 antioxidants-11-00573-f002:**
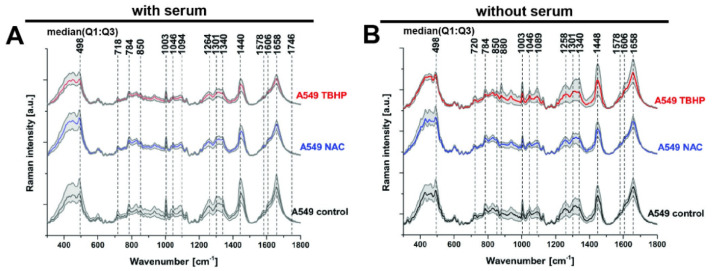
Peak identification in median Raman spectra of A549 live cells treated with either NAC or TBHP, with or without serum. Final concentration of NAC was 1 mM and 200 μM for TBHP. Raman spectra were acquired at 785 nm, with 1 s exposure and 30 accumulation at 120 mW. (**A**) Media with serum: for control and NAC (each n = 610 spectra from 61 cells, 3 biological replicates) and TBHP (n = 810 spectra from 81 cells, 4 biological replicates). (**B**) Media without serum: control and NAC (each n = 400 spectra from 40 cells, 2 biological replicates) and TBHP (n = 610 spectra from 61 cells, 3 biological replicates). Time of the treatments was 1 h. Raman spectra presented as median with first (Q1: 25%) and third quartile (Q3: 75%) (grey background).

**Figure 3 antioxidants-11-00573-f003:**
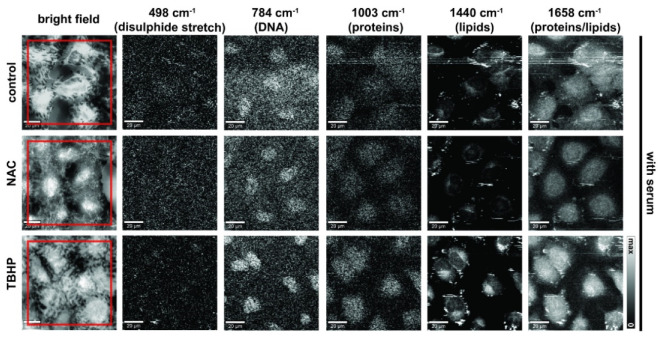
Raman images of live A549 cells in media with serum prepared using different vibrational modes. Bright-field, stitched bright-field microscopy images. Raman images prepared to reflect nucleic acid content (784 cm^−1^, sum filter: 772–796 cm^−1^, scale: 0–150 cts), proteins (498 cm^−1^ (disulphide stretch), sum filter: 475–501 cm^−1^, scale: 0–150 cts and 1003 cm^−1^, sum filter: 991–1015 cm^−1^, scale: 0–150 cts) and lipids/proteins (1440 cm^−1^, sum filter: 1425–1465 cm^−1^, scale: 0–800 cts; and 1658 cm^−1^, sum filter: 1638–1678 cm^−1^; scale: 0–800 cts). Final concentration of NAC was 1 mM and 200 μM for TBHP. Images were acquired at 785 nm, with 0.5 s exposure at 120 mW. Spatial scale bar: 20 microns. Small floating objects (e.g., cell debris or excretion) might generate streak artifacts (horizontal stripes) observed in some images due to the mechanical raster scanning of the Raman microspectroscopy stage.

**Figure 4 antioxidants-11-00573-f004:**
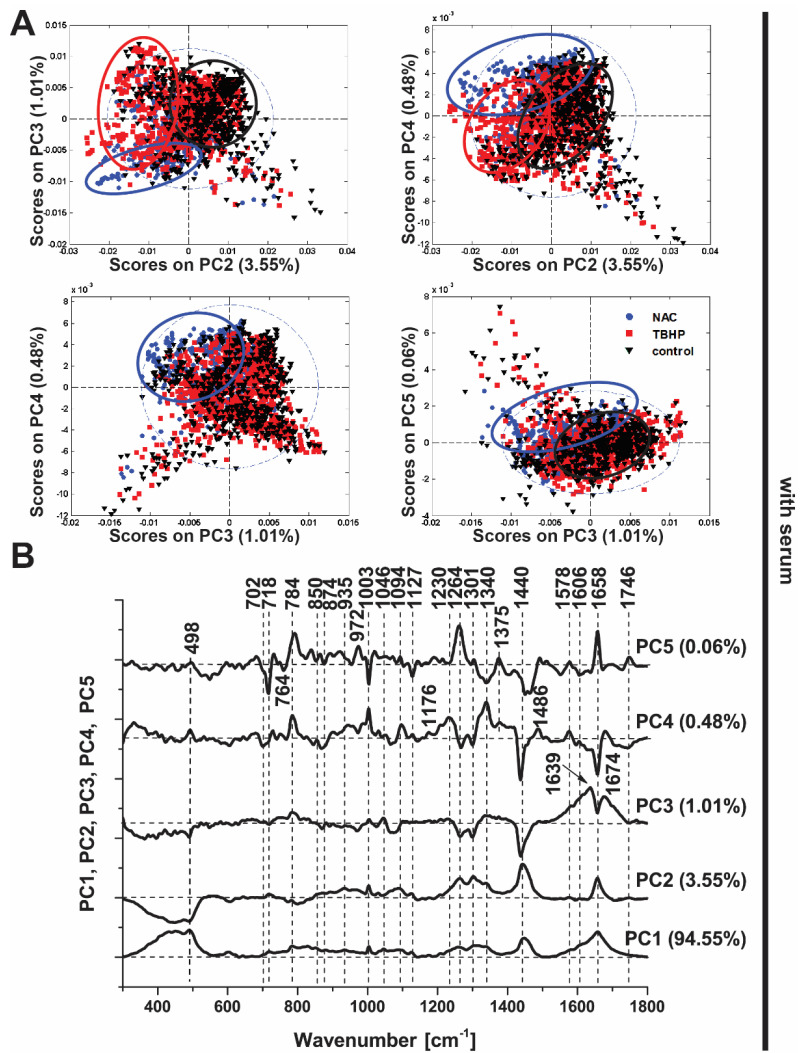
Principal component analysis (PCA) of Raman spectra (culture media with serum). (**A**) PCA scores plots (PC3 vs. PC2, PC4 vs. PC2, PC4 vs. PC3, PC5 vs. PC3). Scores plots show clustering of Raman spectra belonging to the treatment classes: NAC (blue circle), TBHP (red square) and control (black triangle), indicating the potential of Raman spectroscopy to discriminate among the conditions. (**B**) Loadings plot of PC1, PC2, PC3, PC4 and PC5 indicate the Raman bands that contribute to each principal component and hence are responsible for the discrimination between the conditions.

**Figure 5 antioxidants-11-00573-f005:**
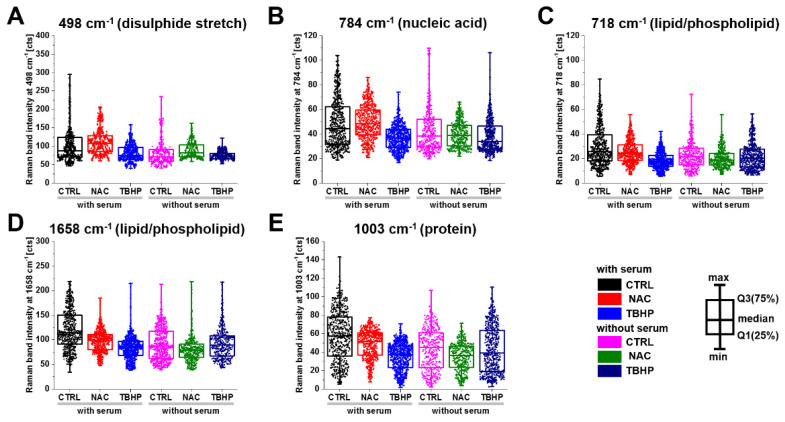
Changes in the Raman band peak intensity of A549 live cells treated with either NAC or TBHP in media with serum. Box plots indicate median (Q1:25%; Q3:75%). The populations were significantly different within and between groups for all presented Raman bands at (**A**) 498 cm^−1^, (**B**) 784 cm^−1^, (**C**) 718 cm^−1^, (**D**) 1658 cm^−1^ and (**E**) 1003 cm^−1^ according to K-W ANOVA at the *p* < 0.05 level (detailed analysis are presented in Appendix A).

**Figure 6 antioxidants-11-00573-f006:**
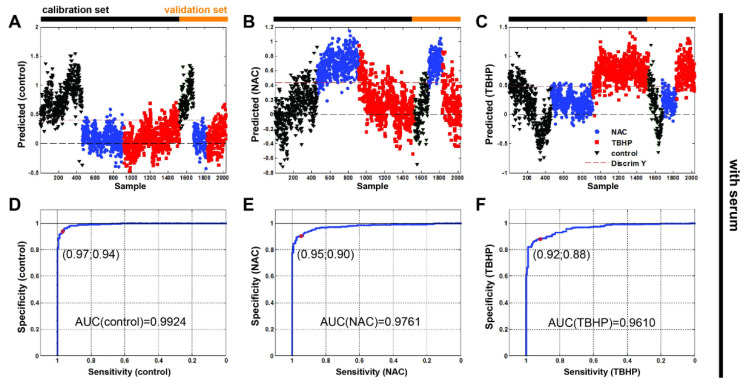
Partial least squares–discriminant analysis with receiver operating characteristic show discrimination ability between the three groups of Raman data for monitoring oxidative stress in vitro in live human cells. The Raman spectra were split into sets for calibration ((**A**) n(control) = 458, (**B**) n(NAC) = 457, (**C**) n(TBHP) = 608) and validation (n(control) = 152, n(NAC) = 153, n(TBHP) = 202; media without serum)) shown left to right, respectively. Control (black triangle), NAC (blue circle) and TBHP (red square). The red dashed line represents discrimination between samples. Receiver operating characteristic (ROC) curves of all Raman data: (**D**) control, (**E**) NAC and (**F**) TBHP.

**Table 1 antioxidants-11-00573-t001:** Raman vibrational mode assignments for the identified bands.

Raman Band Wavenumber (cm^−1^)	Raman Vibrational Mode Assignment
498	S-S disulphide stretching [35,36]
718	CN^+^-(CH_3_)_3_ symmetric stretching, phospholipids [37,38]
784	Cytosine, uracil, thymine, pyrimidine bases, ring breathing modes in DNA bases [37,38,39]
880	Indole ring mode of tryptophan [34]
1003	Phenylalanine, proline, symmetric stretching (ring breathing) mode of phenyl group [36,37]
1094	Symmetric PO_2_^−^ stretching mode of the DNA backbone [38,39]
1264	=CH deformation, triglycerides (fatty acids), lipids [38]
1301	CH_2_ twist, triglycerides (fatty acids), lipids [38]
1440	CH_2_ and CH_3_ deformations, lipids [38]
1606	Tyrosine, phenylalanine ring vibration C=C bending, cytosine NH_2_, protein [36,38]
1658	Amide I, C=O stretching mode, peptide linkage; C=C stretching, lipids [36,38]
1746	C=O stretching, ester group of lipids and phospholipids [38]

**Table 2 antioxidants-11-00573-t002:** Classification results from the partial least squares–discriminant analysis. Abbreviations: RMSEC, room mean square error of calibration; RMSECV, root mean square error of cross-validation; and RMSEP, root mean square error of prediction.

Treatment Condition	Sensitivity	Specificity	RMSEC	RMSECV	RMSEP
Control	96.7	93.5	0.2576	0.2629	0.2465
NAC	94.8	90.1	0.3059	0.3101	0.2930
TBHP	91.6	87.9	0.3262	0.3299	0.3278
Control without serum	89.0	91.7	0.2721	0.2755	0.2784
NAC without serum	97.0	99.6	0.2343	0.2375	0.2409
TBHP without serum	95.4	94.5	0.2744	0.2774	0.2791

## Data Availability

All raw data and code associated with this manuscript will be made available at: https://doi.org/10.17863/CAM.81229 (accessed on 18 February 2022).

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
