# Peer review of "Evaluation of Label-Free Confocal Raman Microspectroscopy for Monitoring Oxidative Stress In Vitro in Live Human Cancer Cells"

_antioxidants, 2022, doi:10.3390/antiox11030573_

Round 1

Reviewer 1 Report

The paper by Surmacki et al. addresses the possible employment of Raman spectroscopy to assess oxidative status in human cells. The methodological approach is exciting, and the data provided in the manuscript is of high value and would serve the Redox Biology Community. It falls nicely into the scope of the Antioxidants journal.

I have only one minor issue and some advice at the same time. The authors did not mention if the pH of NAC was adjusted. It is quite a forgotten issue, which highly impacts the oxidative status of the cell. It would be valuable to add such information in the manuscript text, as it could probably explain findings in figure 1.

Author Response

We would like to thank the reviewer for their positive comments.

Thank you for your interesting question regarding the pH of NAC. In our study, NAC solutions were prepared in medium DMEM/F-12, which contains HEPES for buffering of the cell culture medium at pH 7.2-7.6. Based on prior studies, 15mM HEPES appeared to be sufficient to buffer 1mM NAC, so no further adjustment was made. This has now been explicitly stated in the manuscript and an explanation of the choice of media with buffering system noted. We have also added a comment to the results section relating to Figure 1.

Reviewer 2 Report

This article made an impression. The method described is non-invasive for living cells and allows you to observe changes in the cell instantly. This approach is a way to observe the molecular toponymy of the cell in vital processes.

The article will be of interest to a wide range of readers, especially young scientists starting their careers.

The article is well organized, detailed information interesting only for a narrow specialist is placed in the “Supplementary Data” section, which makes the article easier to read and does not distract the average reader.

I highly recommend accepting an article for publication that increases the rating of the journal.

Author Response

We would like to thank the reviewer for their positive comments.

Reviewer 3 Report

Non-invasive, non-perturbative imaging of oxidative stress does remain a challenge. The reviewer found that the authors have reported the use of confocal Raman microscopy for label-free monitoring of oxidative stress in live bronchial cells. The present manuscript is confocal Raman microscopy for the detection of oxidative stress in live human cancer cells. The reviewer was concerned whether the authors could explain the novelty of the current manuscript compared to the published one, or whether they simply switched different cell lines. Addressing this important issue, this manuscript would merit further consideration.
Because the reviewer's doubt was that the authors did not cite their own work. Surmacki, Jakub M., Isabel Quirós Gonzalez, and Sarah E. Bohndiek. "Application of confocal Raman micro-spectroscopy for label-free monitoring of oxidative stress in living bronchial cells." Biomedical Vibrational Spectroscopy 2018: Advances in Research and Industry. Vol . 10490. International Society for Optics and Photonics, 2018

Author Response

Thank you for raising the question regarding our prior proceedings article from the SPIE Photonics West conference. The proceedings paper presents some very early analysis of oxidative stress in human bronchial epithelial cells, which was not peer reviewed and laid the groundwork for the present article. Compared to the prior study, in the current manuscript we performed both extensive additional experimental work and further analysis including:

  • Fluorescence microscopy validation of the cell culture conditions;
  • Spatial mapping of the distribution of the Raman spectral profiles under pro and anti oxidant conditions;
  • Evaluation of the cellular responses under conditions of serum starvation;
  • Statistical analysis of the Raman band peak intensity for key molecular modifications induced by the pro- and anti-oxidant conditions, including the disulphide stretch, nucleic acids, lipids and phospholipids and proteins;
  • A comprehensive multivariate analysis and modelling showing improved classification potential.

We also performed a careful evaluation of the spectral changes detected in order to assign their relative importance in redox biology and assess the future potential of Raman spectroscopy. We hope that this explanation is sufficient to allay the concerns of the reviewer regarding the novelty of the present study. We have now also referenced the proceedings paper in our introduction for transparency.

Round 2

Reviewer 3 Report

The current manuscript has responded to the reviewer's concern.